# Evaluation of GVPC and BCYE Media for *Legionella* Detection and Enumeration in Water Samples by ISO 11731: Does Plating on BCYE Medium Really Improve Yield?

**DOI:** 10.3390/pathogens9090757

**Published:** 2020-09-16

**Authors:** Maria Scaturro, Elisa Poznanski, Mariarosaria Mupo, Paola Blasior, Margit Seeber, Anna-Maria Prast, Elisa Romanin, Antonietta Girolamo, Maria Cristina Rota, Antonino Bella, Maria Luisa Ricci, Alberta Stenico

**Affiliations:** 1Department of Infectious Diseases, Istituto Superiore di Sanità, Viale Regina Elena 299, 00161 Rome, Italy; maria.scaturro@iss.it (M.S.); antonietta.girolamo@iss.it (A.G.); mariacristina.rota@iss.it (M.C.R.); antonino.bella@iss.it (A.B.); 2Biological Laboratory, Provincial Agency for the Environment and Climate Protection, 39055 Laives (Bolzano), Italy; elisa.poznanski@provincia.bz.it (E.P.); mariarosaria.mupo@provincia.bz.it (M.M.); paola.blasior@provincia.bz.it (P.B.); margit.seeber@provincia.bz.it (M.S.); anna-maria.prast@provincia.bz.it (A.-M.P.); elisa.romanin@provincia.bz.it (E.R.)

**Keywords:** *Legionella*, Legionnaires’ disease, culture, BCYE and GVPC media

## Abstract

*Legionella spp* are the causative agents of Legionnaires’ diseases, which is a pneumonia of important public health concern. Ubiquitous freshwater and soil inhabitants can reach man-made water systems and cause illness. *Legionella* enumeration and quantification in water systems is crucial for risk assessment and culture examination is the gold standard method. In this study, *Legionella* recovery from potable water samples, at presumably a low concentration of interfering microorganisms, was compared by plating on buffered charcoal yeast extract (BCYE) and glycine, vancomycin, polymyxin B, cycloheximide (GVPC) *Legionella* agar media, according to the International Standard Organization (ISO) 11731: 2017. Overall, 556 potable water samples were analyzed and 151 (27.1%) were positive for *Legionella*. *Legionella* grew on both BCYE and GVPC agar plates in 85/151 (56.3%) water samples, in 65/151 (43%) on only GVPC agar plates, and in 1/151 (0.7%) on only BCYE agar plates. In addition, GVPC medium identified *Legionella* species other than *pneumophila* in six more samples as compared with the culture on BCYE. Although the medians of colony forming units per liter (CFU/L) detected on the BCYE and GVPC agar plates were 2500 and 1350, respectively (*p*-value < 0.0001), the difference did not exceed one logarithm, and therefore is not relevant for *Legionella* risk assessment. These results make questionable the need to utilize BCYE agar plates to analyze potable water samples.

## 1. Introduction

*Legionella* is a water-born pathogen widely spread in man-made water systems, responsible for a severe pneumonia and a flu-like illness, named Legionnaires’ disease (LD) and the Pontiac fever, respectively. Overall, at the present time, 62 *Legionella* spp. have been identified and less than a half were pathogenic, however, *Legionella pneumophila* was surely the most frequently found in LD cases. After its first isolation following the large outbreak in Philadelphia in 1976, *Legionella* has become an opportunistic pathogen of major concern, because of a worldwide increasing number of both sporadic cases and outbreak events [1,2,3,4]. Outbreak investigations have widely demonstrated that the most frequent sources of infection are water systems of different buildings, such as hotels or hospitals and, specifically, showers, cooling towers, and spa pools [5,6].

The timely identification of the source of an infection is of great importance to prevent clusters or outbreaks and culture examination is the gold standard for the analyses of water samples. Although molecular methods have been demonstrated to be highly sensitive and specific, as well as able to detect all *Legionella* species and serogroups, they remain impracticable for *Legionella* enumeration because they detect DNA of both living and dead bacteria [7,8].

The *Legionella* laboratory isolation is of great relevance for further deeper molecular investigations, in order to characterize clinical and environmental strains and identify the source of infection [9]. Furthermore, according to the European guidelines for *Legionella* [10], a quantitative evaluation of the contamination of water systems due to *Legionella* can be determined only by a culture, even though a culture has been demonstrated to have some drawbacks [11]. The counting of colony forming units per liter (CFU/L) is a crucial step for risk assessment and, as a consequence, to decide the right control measures to be adopted. The fastidious growth requirements of *Legionella*, the overgrowth of other bacteria, as well as the medium required by the specifically adopted culture method can affect the results of the analysis and determine a variable range of *Legionella* concentrations.

Culture methods are generally performed according to standards, such as the international standard organization (ISO), recognized by each country’s accreditation body [12]. In particular, the ISO 11731 is the most used and it has recently been updated, replacing the previous ones published in 1998 and 2004 (ISO 11731: 2017). Chemical formulation of culture media, as well as the acid and heat treatments, required by the ISO 11731 for the enumeration of *Legionella* in water samples, may have different effects on the recovery of *Legionella*, independent of the manufacturer of the commercially available media [13,14].

The updated ISO 11731 introduced the utilization of the following three media: the buffered charcoal yeast extract (BCYE) agar; the BCYE with selective supplements (BCYE + AB), containing polymixin B, sodium cefazolin, and pimaricin; and the highly selective Modified Wadowsky Yee (MWY) agar or, as an alternative, the glycine, vancomycin, polymyxin B, cycloheximide (GVPC) agar. The choice of the selective medium to be used is linked to the specific potential bacterial contamination of water samples. Potable waters and any other water samples with background microorganisms must be analyzed using selective media, with the capability of reducing background microorganisms. In the ISO 11731: 2017, GVPC and MWY are both considered to be equally efficient for *Legionella* recovery. It has been demonstrated that MWY was the best medium for isolating *Legionella pneumophila* from potable water samples and GVPC was the most effective for reducing additional microbial flora [15,16]. According to ISO 11731: 2017, the decision matrix in Annex J shows that plating on BCYE agar is specifically required for potable water samples when the enumeration of *Legionella* is determined by the following methods: (i) direct plating without any concentration and treatment, (ii) membrane filtration and direct placing of the membrane filter on culture media, and (iii) membrane filtration followed by washing procedure.

In this study, the *Legionella* recovery from potable water samples, concentrated by filtration with washing procedure, was determined and the recovery after plating on BCYE and GVPC agar plates was compared.

## 2. Material and Methods

### 2.1. Water Samples and Culture

Overall, 556 water samples were collected from accommodation sites, hospitals, and private homes and, according to the ISO 11731: 2017, they were classified as belonging to the identified Matrix A, being water samples expected at low concentration of interfering microorganisms. Sampling was performed at different sampling points (shower, faucet, boiler, etc.) of accommodation sites and hospitals according to protocols reported in Italian guidelines [17]. Briefly, an instant water sample was collected to simulate exposure by a user, without flaming and disinfecting the outlet, and without running water. The temperature was measured immediately before filling the one-liter bottle.

The BCYE and GVPC (Oxoid, Thermo Fisher Diagnostics Limited, Cheshire, UK) agar plates and reagents were prepared according to ISO 11731: 2017. Legionella CYE agar base, BCYE-α growth supplement, and GVPC selective supplement were purchased from Oxoid, Thermo Fisher Diagnostics Limited, Cheshire, UK. For each lot of both media, a quality control was carried out according to ISO 11133: 2014 [18]. The reference material Easy-tab Reference Material (LGC, Bury, UK) was utilized for performance testing of the two media, and always resulted within the declared range. Selectivity of GVPC was qualitative determined according to ISO 11133: 2014. Ringer solution was used to wash polycarbonate membranes using a vortex mixer.

For each sample, a volume of one liter was collected and it was concentrated 200 times by filtration, using 0.22 μm polycarbonate membrane, followed by the washing procedure of the filter. After filtration the membrane was placed in a screw cap sterile container with 5 mL of diluted Ringer’s solution. The membrane was washed by shaking vigorously for at least 2 min using a vortex mixer. The concentrated sample was divided into the following three aliquots, according to ISO 11731: 2017: one ml was heat treated, one ml was acid treated, and the remaining 3 mL were untreated. Then, one hundred μL of each aliquot were placed on both BCYE and GVPC agar plates and incubated at 37 °C, for ten days. The plates were checked after four or five days, and after 10 days. According to the concentration procedure, the detection limit of our method was 50 CFU/L.

The laboratory that analyzed the samples is accredited for the detection and enumeration of *Legionella* according to ISO 11731: 2017, by the Italian national accreditation body (Accredia).

### 2.2. Statistical Analysis

The McNemar’s test was used to compare frequency on paired data. The concordance between media was evaluated using the Kappa test (K < 0.20 = “poor”, 0.20–0.40 = “fair”, 0.40–0.60 = “moderate”, 0.60–0.80 = “good”, and 0.80–1.00 = “very good”). Specificity and sensitivity, as well positive and negative predictive values (PPV and NPV, respectively), and 95% confidence intervals (CI) for both media were calculated, considering the BCYE as a reference medium.

All statistical analyses were performed by Stata software version 11.2 (Stata Corp, College Station, TX, USA).

## 3. Results 

All of the 556 water samples were analyzed by culture and, overall, 151 (27.1%) were positive for Legionella, of which 65 (43%) grew on only GVPC, 85 (56.3%) grew on both GVPC and BCYE, and one (0.7%) grew on only BCYE. (Table 1). The difference of the results obtained analyzing the samples by the two media was significant (McNemar’s test, *p* < 0.0001).

The sensitivity, specificity, positive predictive value (PPV), and negative predictive value (NPV) of GVPC vs. BCYE media are shown in Table 2. GVPC demonstrated a greater sensitivity and a good specificity as compared with BCYE medium. The Cohen’s Kappa coefficient calculated on these data provided a value of 0.65, indicating a good quality of agreement (*p* < 0.0001) (Table 2).

Considering the 85 samples that were positive on both GVPC and BCYE media, 11 samples showed fewer colony forming units per liter (CFU/L) on GVPC than on BCYE (Table 3). However, the differences of CFU/L found between the two media were never higher than 1 log (*p*-value = 0.0388).

Figure 1 shows the distribution of *L. pneumophila* serogroups and *Legionella* species found on GVPC and BCYE agar plates. *Legionella* species were detected in 19 samples by GVPC and in 13 samples by BCYE. Considering the range of the colony forming unit (CFU)/L detected, the data proved that, among the 65 samples positive on only GVPC, there were 51 samples (78.4%) showing a range between 50 and 1000, highlighting the high efficiency of GVPC in isolating low bacterial counts. The calculated medians of CFU/L detected on BCYE and GVPC plates were equal to 2500 (interquartile range = 5500) on BCYE and 1350 (interquartile range = 3950) on GVPC, and the difference between the two media was significant (*p*-value < 0.0001).

## 4. Discussion

In this study, potable water samples were analyzed according to the ISO1173: 2017, in order to compare the *Legionella* recovery obtained by plating on BCYE and GVPC agar plates. We observed that GVPC was more efficient in detecting *Legionella* than BCYE medium. Indeed, 43% of the overall positive samples were detected on only GVPC agar plates and, in addition, the positivity of water samples at low bacterial counts, corresponding to 78.4% of the total positive samples, was determined only by using this medium. This finding has significant relevance especially when, in specific water systems as hospital wards or thermal waters, the absence or a strong containment of Legionella contamination must be guaranteed, due to possible exposure by people at increased risk of acquiring LD. In a previous study, it was demonstrated that a much greater yield of *Legionella* spp. was obtained by plating on BCYE than on MWY agar plates, and a significantly higher number of CFU of both *Legionella pneumophila* and non-pneumophila was counted on BCYE as compared with MWY [19]. On the contrary, Leoni et al. demonstrated the significantly higher yield on GVPC and MWY than on BCYE medium, in combination with the technique used of direct inoculum or pretreatment with acid or heat [18]. Furthermore, other studies have shown no statistically significant differences between BCYE and GVPC media in recovering *Legionella* in water samples [14,20]. In this study, *Legionella* recovery was determined by comparing the CFU/L counted in BCYE and GVPC, and although significantly higher CFU/L were found in BCYE than in GVPC agar plates, the difference did not exceed one log. Therefore, it was not relevant for *Legionella* risk assessment of drinking water systems.

Furthermore, the possibility of improving the isolation of *Legionella* non-*pneumophila* species by plating the water samples on BCYE was not confirmed. Indeed, *Legionella* species other than *pneumophila* were detected more on GVPC than on BCYE.

In conclusion, these results cast doubt on the advantages of analyzing water samples using only BCYE, as required by ISO 11731: 2017. Further investigations by analyzing a larger number of water samples should be conducted to confirm these data, which, if confirmed, would bring enormous benefits, saving time and money, especially for laboratories that carry out monitoring activities and analyze hundreds of samples daily.

## Figures and Tables

**Figure 1 pathogens-09-00757-f001:**
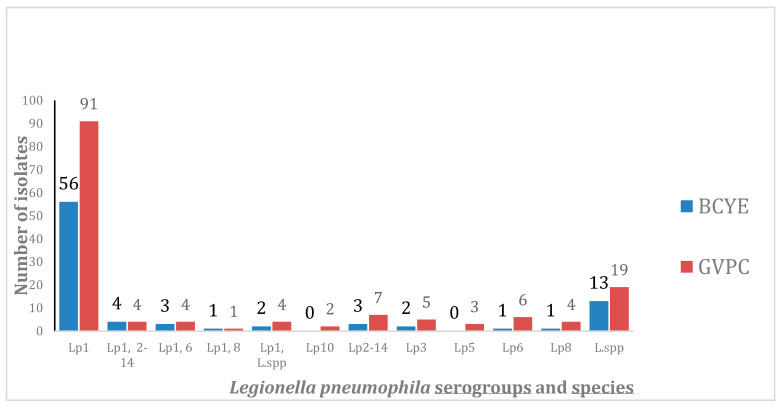
Distribution of *Legionella pneumophila* serogroups and *Legionella* species (BCYE n = 86 vs. GVPC n = 150).

**Table 1 pathogens-09-00757-t001:** Recovery of *Legionella* by using buffered charcoal yeast extract (BCYE) and glycine, vancomycin, polymyxin B, cycloheximide (GVPC) media.

	GVPC			
**BCYE**
		**Negative**	**Positive**	**Total**
	Negative	405	65	470
	Positive	1	85	86
	Total	406	150	556

McNemar’s test *p*-value <0.0001.

**Table 2 pathogens-09-00757-t002:** GVPC vs. BCYE sensitivity, specificity, positive predictive value (PPV), negative predictive value (NPV), concordance, Kappa value, and *p*-value.

Comparison	% (95% CI)
Sensitivity	98.8 (97.9–99.7)
Specificity	86.2 (83.3–89.0)
PPV	56.7 (52.5–60.8)
NPV	99.7 (99.3–100.2)
Concordance	88.1
Kappa value (*p*-value)	0.65 (<0.0001)

CI, confidence interval; PPV, positive predictive value; NPV, negative predictive value.

**Table 3 pathogens-09-00757-t003:** *Legionella* CFU/L range detected on samples positive on both BCYE and GVPC media.

		GVPC			Total
	CFU/L	50–1000	1050–10,000	>10,000	
**BCYE**	50–1000	30	3	0	33
	1050–10,000	5	33	0	38
	>10,000	0	6	8	14
	Total	35	42	8	85

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
