# Peer review of "Evaluation of GVPC and BCYE Media for Legionella Detection and Enumeration in Water Samples by ISO 11731: Does Plating on BCYE Medium Really Improve Yield?"

_pathogens, 2020, doi:10.3390/pathogens9090757_

Round 1
Reviewer 1 Report
This is a well written and interesting paper and worthy of publication and further work. I think you need to include the limit of detection for each media type as I understand that in waters with >1000cfu/L it makes very little difference but my concern would be in water samples of 10-100 cfu/L if solely relying on GVPC would these have been missed. I understand they are not alert levels but it is important that we can still detect low levels as it is an early warning system for potential problems within a water system particularly within high risk areas like hospitals. This study could be extended Europe wide quite easily as many water labs will be following the ISO 11731:2017 method and only with greater numbers and evidence could the ISO 11731:2017 perceivable be changed.
Author Response
Question. I think you need to include the limit of detection for each media type as I understand that in waters with >1000cfu/L it makes very little difference but my concern would be in water samples of 10-100 cfu/L if solely relying on GVPC would these have been missed. I understand they are not alert levels but it is important that we can still detect low levels as it is an early warning system for potential problems within a water system particularly within high risk areas like hospitals. This study could be extended Europe wide quite easily as many water labs will be following the ISO 11731:2017 method and only with greater numbers and evidence could the ISO 11731:2017 perceivable be changed. Answer Our detection limit for both media is 50 CFU/L according to the concentration procedure. The recovery of Legionella in the two culture media was performed by using of Easy-tab Reference Material (LGC, UK) resulting always within the declared range.
Reviewer 2 Report
Line 83
Please explain further the sampling procedure (eg sampling points, the existence of cooling towers … etc)
Lines 92-97
Please give more information about the filtration process
(eg after filtration the membrane was placed in a sterile beaker or stomacher bag with certain ml of distilled water or ringer solution……. etc)
The three aliquots produced were of how many ml ?
Was there any centrifugation involved in order to produce the final aliquots ?
Aliquots placed on both BCYE and GVPC plates were plated on the surface of each agar plate ? (I suppose so but please make it clear)
Lines 107-113 and Table 1
Please rephrase the whole paragraph for easier understanding of numbers, one has to read several times the paragraph and the table and make calculations on paper in order to find the 151 positive samples on both media. The use of a flow diagram instead of a table could facilitate the depiction of +ve and –ve numbers quickly.
Line 141
Please explain with more details how the 78.4% was estimated
Author Response
Reviewer 2
Line 83
Question. Please explain further the sampling procedure (eg sampling points, the existence of cooling towers … etc). Answer. Sampling was performed at different sampling points (shower, faucet, boiler, etc) of accommodation sites and hospitals according to protocols reported in Italian guidelines. Briefly, an instant water sample was collected to simulate exposure by a user, without flaming and disinfecting the outlet, and without running water. The temperature was measured immediately before filling the one-liter bottle.
Lines 92-97
Question. Please give more information about the filtration process(eg after filtration the membrane was placed in a sterile beaker or stomacher bag with certain ml of distilled water or ringer solution……. etc). Answer. After filtration the membrane was placed in a screw cap sterile container with 5 ml of diluted Ringer’s solution. The membrane was washed by shaking vigorously for at least 2 min using a vortex mixer.
Question The three aliquots produced were of how many ml ? Answer .The concentrated samples were divided in three aliquots subjected to different treatments (according to ISO 11731:2017): one ml was heat-treated, one ml was acid-treated and the remaining 3 ml were untreated.
Question. Was there any centrifugation involved in order to produce the final aliquots? Answer. No centrifugation step was applied to the samples.
Question Aliquots placed on both BCYE and GVPC plates were plated on the surface of each agar plate? (I suppose so but please make it clear). Answer. One -hundred microliters were plated on both BCYE and GVPC agar plates as already mentioned (line 106)
Lines 107-113 and Table 1
Question. Please rephrase the whole paragraph for easier understanding of numbers, one has to read several times the paragraph and the table and make calculations on paper in order to find the 151 positive samples on both media. The use of a flow diagram instead of a table could facilitate the depiction of +ve and –ve numbers quickly. Answer. All the 556 water samples were analysed by culture and overall 151 (27.1%) were positive for Legionella, of which 65(43%) grew only on GVPC; 85(56.3%) grew on both GVPC and BCYE and only 1 (0.7%) grew on BCYE. (Table1). The difference of the results obtained analysing the samples by the two media was significant (McNemar’s test: p<0.0001).
Line 141
Question. Please explain with more details how the 78.4% was estimated.
Answer. Considering the range of the of colony forming unit (CFU)/L detected, the data proved that among the 65 samples positive only on GVPC there were 51 (78.4%) showing a range between 50 and 1000, highlighting the high efficiency of GVPC in isolating low bacterial counts.